# Augmented Reality Navigation System (SIRIO) for Neuroprotecion in Vertebral Tumoral Ablation

**Eliodoro Faiella, Rebecca Casati, Matteo Pileri \*** , **Giuseppina Pacella, Carlo Altomare, Elva Vergantino** , **Amalia Bruno, Bruno Beomonte Zobel and Rosario Francesco Grasso**

Department of Diagnostic and Interventional Radiology, University Hospital Campus Bio-Medico of Rome, Via Alvaro del Portillo, 200, 00128 Rome, Italy; e.faiella@policlinicocampus.it (E.F.); rebecca.casati@unicampus.it (R.C.); g.pacella@policlinicocampus.it (G.P.); c.altomare@policlinicocampus.it (C.A.); elva.vergantino@unicampus.it (E.V.); amalia.bruno@unicampus.it (A.B.); b.zobel@policlinicocampus.it (B.B.Z.); r.grasso@policlinicocampus.it (R.F.G.)

\* Correspondence: matteo.pileri@unicampus.it

**Abstract:** (1) This study evaluates the impact of the CT-guided SIRIO augmented reality navigation system on the procedural efficacy and clinical outcomes of neuroprotection in vertebral thermal ablation (RTA) for primary and metastatic bone tumors. (2) Methods: A retrospective non-randomized analysis of 28 vertebral RTA procedures was conducted, comparing 12 SIRIO-assisted and 16 non-SIRIO-assisted procedures. The primary outcomes included dose-length product (DLP) and epidural dissection time. The secondary outcomes included technical success, complication rates, and pain scores at procedural time (VAS Time 0) and three months post-procedure (VAS Time 1). The statistical analyses included t-tests, Mann–Whitney U tests, and multiple regression. (3) Results: SIRIO-assisted procedures significantly reduced DLP (307.42 mGycm vs. 460.31 mGycm, $p = 2.23 \times 10^{-8}$) and procedural epidural dissection time (13.48 min vs. 32.26 min, $p = 2.61 \times 10^{-12}$) compared to non-SIRIO-assisted procedures. Multiple regression confirmed these reductions were significant (DLP: $\beta = -162.38$, $p < 0.001$; time: $\beta = -18.25$, $p < 0.001$). Pain scores (VAS Time 1) did not differ significantly between groups, and tumor type did not significantly influence outcomes. (4) Conclusions: The SIRIO system enhances neuroprotection efficacy and safety, reducing radiation dose and procedural time during spine tumoral ablation while maintaining consistent pain management outcomes.

**Keywords:** interventional oncology; neuroprotection techniques; epidural dissection; navigation system; radiation dose; bone ablation; radiofrequency

## 1. Introduction

The progress of interventional oncology has led to the development of sophisticated technologies aimed at improving the precision and results of procedures such as vertebral thermal ablation (RTA). Vertebral RTA coupled with neuroprotection has become a critical treatment mode for patients with primary and metastatic bone tumors, in particular for the treatment of the lesions of the rear vertebral wall, near neural structures. The main objectives of these interventions are to alleviate pain, stabilize the affected areas, and minimize the risk of neurological complications [1].

To avoid damage to the neighboring neural structures during treatment, which typically occurs at temperatures above 45 °C or lower than 10 °C, different neuroprotection techniques can be used, such as passive monitoring, active protection, or both [2].

Passive monitoring involves the use of thermosensors for the direct temperature monitoring of a risk structure. On the other hand, the active protection techniques include pneumo-dissection (with carbon dioxide or ambient air), hydro-dissection (with saline solution, contrast or dextrose means), or hydro-convery (a constant flow of protective fluid) in epidural space, in an attempt to provide a thermal barrier to the neuronal structure.

The introduction of the CT-guided infra-red augmented reality navigation system SIRIO (MASMEC S.p.A., Modugno, Bari, Italy) has revolutionized the approach to these complex procedures [3]. The SIRIO Augmented Reality Navigation System is an intraoperative tool that reconstructs a 3D model from CT images using a semi-automatic algorithm. It consists of a patient tool (PT), a needle tool (NT), a visualization unit (VU), and an infrared optical sensor (OS). The PT is positioned near the target area to minimize movement, while the NT is attached to the biopsy needle or the ablation tool. Infrared light reflected by the spheres on the PT and NT is detected by the OS. A proprietary algorithm analyzes the CT images to create a 3D model, which is aligned with the patient's anatomy through automatic calibration. During the procedure, the axial and sagittal projections of the model are displayed on the VU and updated in real time based on the position of the NT. This allows the tool to be guided into the lesion with extreme precision. Despite the intuitive advantages of this advanced technology, comprehensive clinical data demonstrating its real-world effectiveness remain limited. The SIRIO navigation system is designed to improve the accuracy of probe positioning, optimize procedural workflows, and reduce radiation exposure for both patients and healthcare providers [4]. Previous studies have explored various navigation systems for improving the precision of vertebral ablation procedures [5–7]. Similarly, other studies highlighted the potential of low-dose CT-guided navigation systems in reducing radiation exposure during bone ablations [4]. However, these studies have not fully addressed the impact of such systems on neuroprotection in vertebral ablation procedures.

This study aims to address this gap by evaluating the impact of the SIRIO navigation system on procedural efficacy and clinical outcomes in vertebral RTA with neuroprotection. Specifically, we aim to determine if SIRIO-assisted procedures result in reduced dose-length product (DLP) and shorter procedural times compared to non-SIRIO-assisted procedures. A secondary objective is to assess the system's impact on pain management outcomes.

## 2. Materials and Methods

This retrospective study analyzed 28 vertebral thermal ablation (RTA) procedures performed between September 2020 and March 2023. The study compared outcomes between SIRIO-assisted ($n = 12$) and non-SIRIO-assisted ($n = 16$) procedures.

Inclusion criteria were age > 18 years, vertebral lesions (metastases or osteoid osteomas) abutting the posterior vertebral somatic wall, and a clinical indication for RFA. Exclusion criteria included contraindications for percutaneous interventions, refusal to provide written informed consent, and poor patient compliance, according to CIRSE guidelines [1]. Informed consent was obtained from all subjects enrolled in the study. The study was conducted in accordance with the Declaration of Helsinki and approved by the Institutional Review Board of Campus Bio-Medico University Hospital.

In the SIRIO-assisted group, the SIRIO augmented reality navigation system was used to guide the needle into the lesion with real-time 3D visualization. Standard CT guidance was employed in the non-SIRIO group. Epidural dissection was performed in all procedures through a 22 G needle, using non-ionic 5% dextrose mixed with diluted organ iodine contrast medium and carbon dioxide (Figures 1 and 2), to obtain a multilevel dissection adjacent to the site to be treated. The procedures were performed under similar conditions, with a focus on minimizing radiation exposure and ensuring precision in needle placement. For each patient, medical records, previous imaging exams, laboratory studies, and lesion-related pathological information were carefully evaluated.

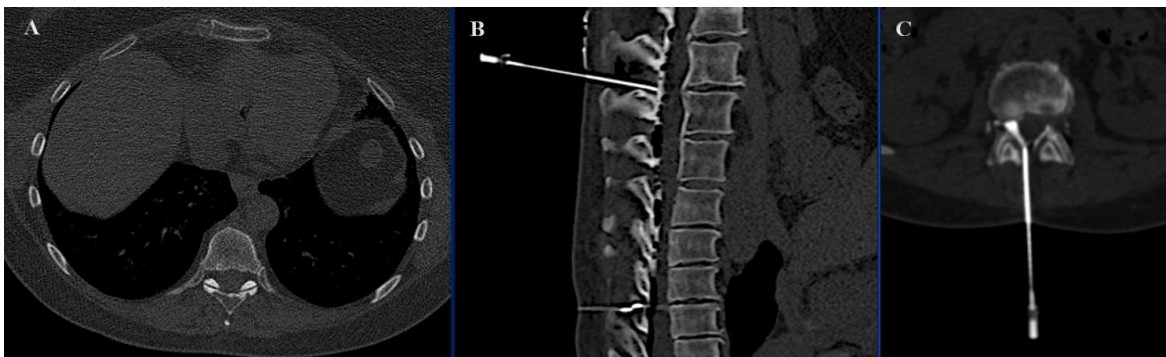

**Figure 1.** Axial (**A**) image demonstrating a secondary osteolytic lesion from breast cancer located in the spinal process of D8. Sagittal (**B**) and axial (**C**) images showing epidural dissection with $CO_2$ and contrast medium.

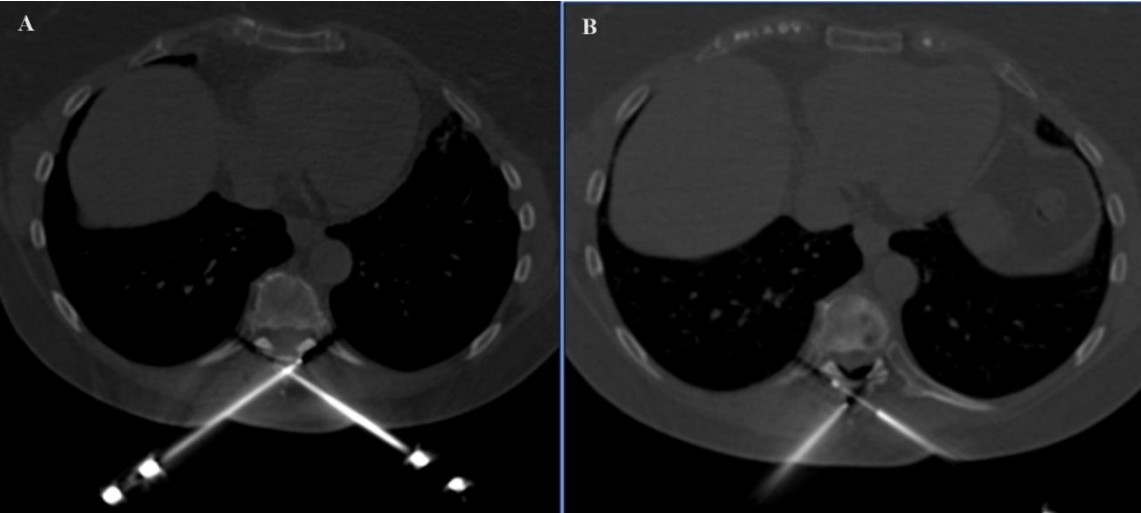

**Figure 2.** Axial images demonstrating the positioning of two probes in the spinal process of D8 (**A**) and the adjacent epidural dissection obtained (**B**).

The primary endpoints were procedural time and patient radiation dose (DLP). Secondary endpoints included technical success, complication rates, and pain scores:

- Procedural Time: Defined as the duration from the initial needle insertion to the completion of optimal dissection for safe ablative treatment;
- Patient Radiation Dose: Measured by the dose-length product (DLP) recorded during the neuroprotection procedure;
- Technical success: Percentage of tumors successfully treated, resulting in a complete ablation zone covering the target tumor, as depicted on immediate post-ablation imaging and first follow-up imaging (1 month);
- Complication Rates: Any adverse events occurring during or after the procedure were monitored and recorded;
- Pain Scores: Measured by the Visual Analog Scale (VAS) at procedural time (VAS Time 0) and three months post-procedure (VAS Time 1).

To evaluate ablation zone dimensions and detect any post-procedural complications, immediate and 24 h CT controls were performed. In the absence of symptoms or other indications, follow-up of treated metastases was usually conducted with contrast-enhanced MRI within 3 months after the treatment or with PET-CT at least 6 months after the treatment to allow inflammation to subside. In this study, an additional follow-up visit was conducted in the interventional radiology outpatient clinic at 3 months. Follow-up at 1 year was performed based on new symptoms or at the discretion of the attending physician.

### 3. Statistical Analysis

A comprehensive statistical analysis was conducted to compare outcomes between SIRIO-assisted and non-SIRIO-assisted procedures using IBM SPSS Statistics (version 28.0.0., IBM Corp., Armonk, NY, USA). Descriptive statistics, including mean, standard deviation, median, and range, summarized primary and secondary outcomes such as DLP, procedural time, technical success, complication rates, and VAS scores. A *p*-value of <0.05 was considered statistically significant.

To determine the significance of differences between the two groups, inferential statistical analyses were performed. The Shapiro–Wilk test assessed data normality. For normally distributed variables, independent samples t-tests compared group means. For non-normally distributed variables, the Mann–Whitney U test was used.

Multiple regression analyses were conducted to understand factors influencing DLP and procedural time. Independent variables included age, tumor type, and SIRIO assistance. These models isolated the effect of each predictor, indicating the direction and magnitude of their impact.

Correlation analyses explored relationships between continuous variables, such as lesion dimensions and VAS scores, using Pearson and Spearman coefficients based on data normality. Partial regression plots visualized individual predictor effects while controlling for other variables.

Subgroup analyses evaluated potential differences across demographic and clinical subgroups, categorizing patients into age groups (<50, 50–70, >70) to explore age-related variations. Statistical comparisons were conducted using ANOVA or the Kruskal–Wallis test, followed by post-hoc pairwise comparisons. The data presented in this study are available on request from the corresponding author.

### 4. Results

Statistical analysis provided robust evidence of the efficacy of the SIRIO navigation aid in neuroprotection procedures for the treatment of vertebral bone tumors. Descriptive statistics clearly delineated differences in both the dose-length product (DLP) and procedural time between procedures facilitated by the SIRIO system and those without its assistance. The average DLP was significantly lower in the SIRIO group, at 307.42 mGycm, compared to 460.31 mGycm in the non-SIRIO group. Similarly, the procedural time was markedly shorter in the SIRIO-assisted group, with a mean of 13.48 min compared to 32.26 min in the non-assisted group. The descriptive statistics further showed the impact of these interventions on pain scores measured by the VAS at procedural time (VAS Time 0) and three months post-procedure (VAS Time 1), with both groups demonstrating substantial pain reduction (Table 1).

Inferential statistical analysis confirmed these findings. As summarized in Table 2, Mann–Whitney U tests revealed significant reductions in both DLP (U = 15.0, *p* = 0.000186) and procedural time (U = 12.0, *p* = 0.000105) with SIRIO assistance. The t-test for VAS Time 1 indicated no significant difference between the groups (t = 0.484, *p* = 0.6321).

Figure 3 illustrates the comparison of DLP (A), procedural time (B), and VAS (C, D) between SIRIO-assisted and non-assisted procedures. The boxplots show a marked reduction in both DLP and procedural time with the use of the SIRIO navigation system, confirming its efficacy in reducing both radiation dose and procedural time, with no significant difference in terms of VAS Time 0 and VAS Time 1 between the two groups.

**Table 1.** Descriptive statistics for the SIRIO and non-SIRIO groups, including Dose-Length Product (DLP), procedural time, needle path, dimension of the treated lesion, VAS 0, and VAS 1. The SIRIO group showed lower DLP, shorter procedural times, and other notable differences compared to the non-SIRIO group.

| Group | Metric | Mean | Min | Median | Max |
|---|---|---|---|---|---|
| SIRIO | DLP (mGycm) | 307.42 ± 48.89 | 221 | 304 | 384 |
| | Procedural Time (min) | 13.48 ± 3.62 | 7.8 | 14.4 | 18.7 |
| | Needle path (mm) | 46.33 ± 6.12 | 36 | 47 | 55 |
| | Dimension of the lesion (mm) | 11.67 ± 8.05 | 6 | 8 | 28 |
| | VAS 0 | 0.75 ± 0.45 | 0 | 1 | 1 |
| | VAS 1 | 0.25 ± 0.45 | 0 | 0 | 1 |
| Non-SIRIO | DLP (mGycm) | 460.31 ± 32.82 | 402 | 457 | 526 |
| | Procedural Time (min) | 32.26 ± 4.44 | 27.1 | 32.5 | 35.6 |
| | Needle path (mm) | 53.00 ± 3.23 | 49 | 54 | 56 |
| | Dimension of the lesion (mm) | 23.50 ± 9.65 | 6 | 25 | 29 |
| | VAS 0 | 0.69 ± 0.48 | 0 | 1 | 1 |
| | VAS 1 | 0.31 ± 0.48 | 0 | 0 | 1 |

**Table 2.** Summary of Inferential Statistics: This table summarizes the results of the inferential statistical tests performed to compare key metrics between SIRIO-assisted and non-SIRIO-assisted procedures.

| Metric | Test Used | Statistic | *p*-Value |
|---|---|---|---|
| DLP (mGy·cm) | Mann–Whitney U | 15.0 | 0.000186 |
| Procedural Time (min) | Mann–Whitney U | 12.0 | 0.000105 |
| VAS Time 0 | Mann–Whitney U | 86.0 | 0.643494 |
| VAS Time 1 | Mann–Whitney U | 0.484 | 0.6321 |

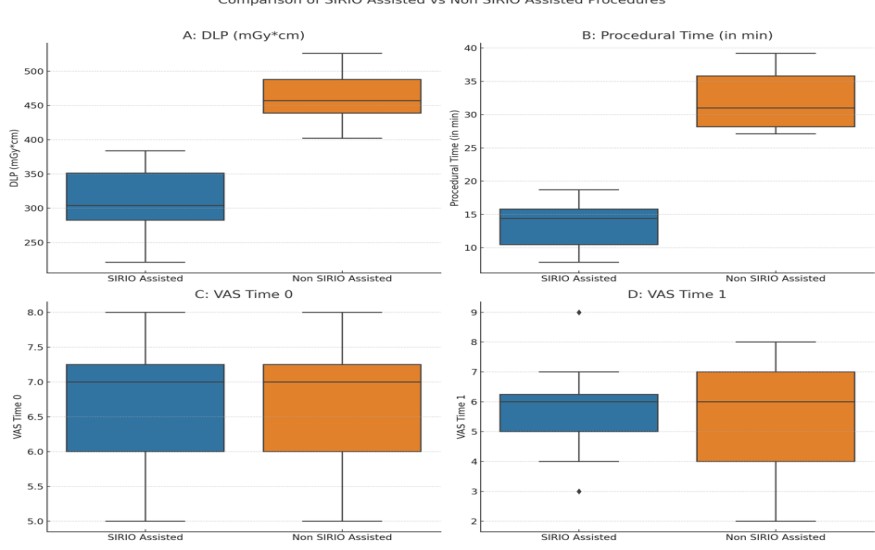

**Figure 3.** Comparison of SIRIO Assisted vs. non-SIRIO Assisted epidural dissections: (**A**): Comparison of Dose-Length Product (DLP) between SIRIO-assisted and non-SIRIO-assisted procedures. The boxplot shows significantly lower DLP values for SIRIO-assisted procedures, indicating reduced radiation exposure. (**B**): Comparison of Procedural Time between SIRIO-assisted and non-SIRIO-assisted procedures. The boxplot indicates that procedural time is significantly shorter for SIRIO-assisted procedures. (**C**): Comparison of Procedural Pain (VAS Time 0) between SIRIO-assisted and non-SIRIO-assisted procedures. The boxplot suggests no significant difference in immediate procedural pain between the two groups. (**D**): Comparison of Pain at Three Months (VAS Time 1) between SIRIO-assisted and non-SIRIO-assisted procedures. The boxplot shows no significant difference in pain reduction at three months post-procedure between the two groups.

Multiple regression analysis, as shown in Figure 4, was conducted to assess the impact of various predictors on DLP and procedural time, including age, tumor type, and SIRIO assistance. The analysis revealed that SIRIO assistance had the most significant negative impact on both DLP (β = −162.38, $p < 0.001$) and procedural time (β = −18.25, $p < 0.001$), indicating substantial efficiency gains. Age also had a significant positive impact on DLP (β = 1.19, $p = 0.014$), although its effect on procedural time was not significant.

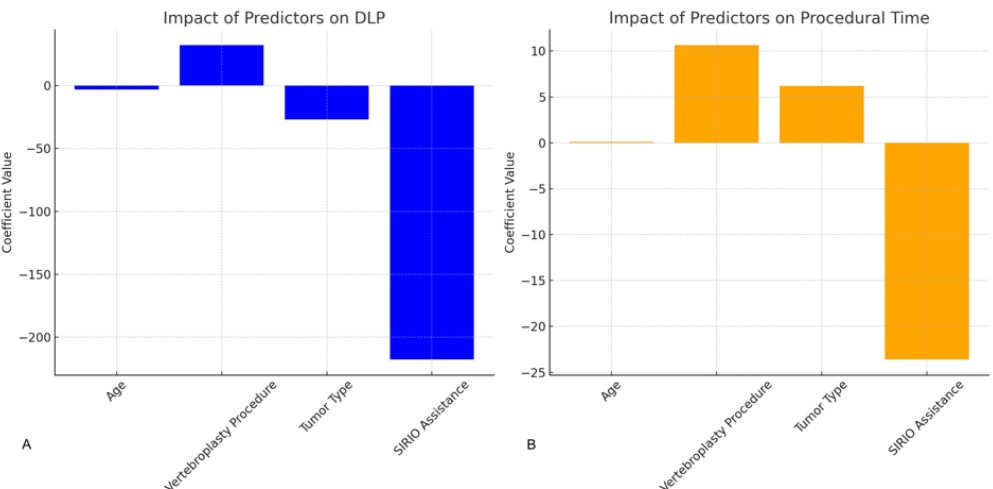

**Figure 4.** Impact of Predictors on DLP and Procedural Time. (**A**): Bar chart showing the impact of predictors (Age, Vertebroplasty Procedure, Tumor Type, SIRIO Assistance) on DLP. The SIRIO Assistance has the most significant negative impact on DLP. (**B**): Bar chart showing the impact of predictors (Age, Vertebroplasty Procedure, Tumor Type, SIRIO Assistance) on Procedural Time. SIRIO Assistance significantly reduces the procedural time.

The relationship between the dimension of the treated lesion and VAS Time 1 for both SIRIO-assisted and non-assisted procedures was explored, as shown in Figure 5. The scatter plots with regression lines show a positive correlation between lesion size and VAS Time 1, with larger lesions associated with higher pain scores at three months post-procedure.

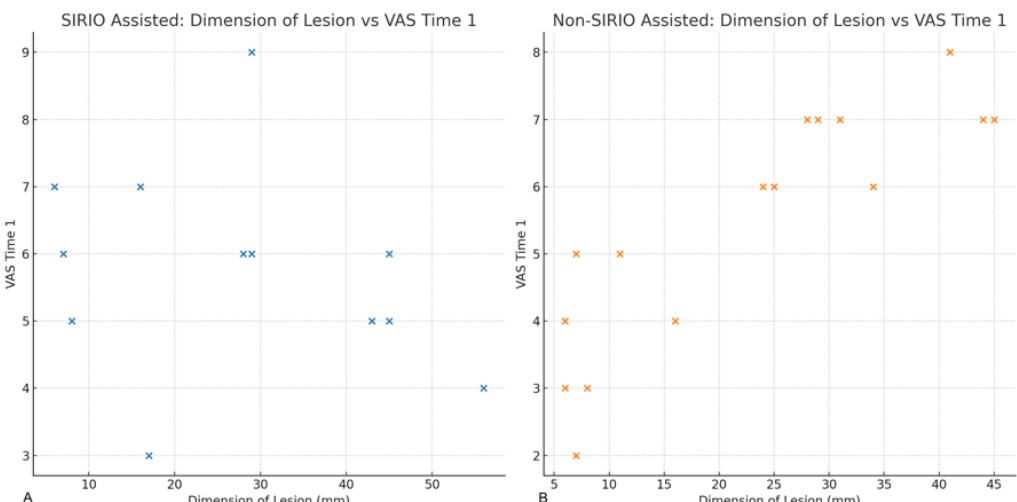

**Figure 5.** Relationship between Dimension of Lesion and VAS Time 1. (**A**): Scatter plot showing the relationship between the dimension of the treated lesion and the VAS score at three months post-procedure for SIRIO-assisted procedures. A positive correlation is observed, indicating that larger lesions tend to be associated with higher pain scores. (**B**): Scatter plot showing the relationship between the dimension of the treated lesion and the VAS score at three months post-procedure for non-SIRIO assisted procedures. A similar positive correlation is observed, suggesting that larger lesions are associated with higher pain scores in both groups.

Subgroup analysis further examined the differences in VAS Time 1 across different age groups for both SIRIO-assisted and non-assisted procedures, as depicted in Figure 6. The boxplots indicate significant differences in VAS scores between age groups in the non-SIRIO-assisted procedures, particularly between the <50 and 50–70 age groups and <50 and >70 age groups, suggesting that younger patients experience less pain post-procedure.

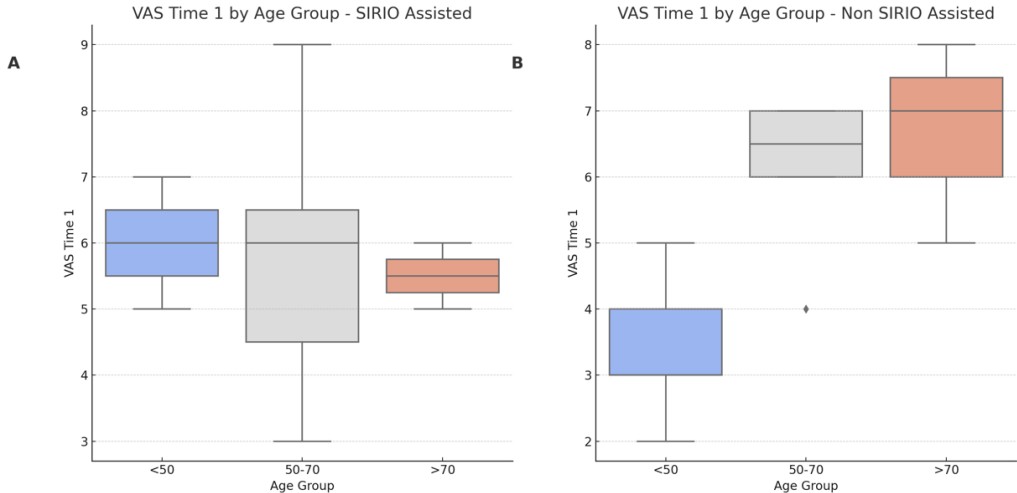

**Figure 6.** VAS Time 1 by Age Group for SIRIO Assisted and Non-SIRIO Assisted Procedures. (**A**): Boxplot showing the distribution of VAS Time 1 scores across different age groups for SIRIO-assisted procedures. No significant differences were found between the age groups. (**B**): Boxplot showing the distribution of VAS Time 1 scores across different age groups for non-SIRIO assisted procedures. Significant differences were found between the age groups, particularly between <50 and 50–70, and <50 and >70.

A Spearman correlation analysis was conducted to explore the relationship between tumor type and VAS Time 1, revealing a very weak positive correlation ($\rho = 0.198$, $p = 0.314$). This suggests that the type of tumor does not significantly influence the pain scores three months post-procedure.

The findings from this study provide compelling evidence that the SIRIO navigation system significantly reduces both the radiation dose (DLP) and procedural time in neuroprotection procedures. The benefits of the SIRIO system are consistent across different patient demographics and tumor characteristics, as demonstrated by the robustness of the results from multiple regression and subgroup analyses. These results underscore the value of integrating advanced navigation systems like SIRIO in enhancing the safety and efficacy of interventional oncology procedures.

## 5. Discussion

Vertebral thermal ablation is a crucial procedure for treating spinal tumors, where precision and safety are paramount, particularly to avoid damage to adjacent neural structures.

The SIRIO system is particularly advantageous in complex procedures requiring precise needle placement, such as vertebral thermal ablation near critical neural structures. Its real-time 3D navigation significantly enhances procedural accuracy, reducing the risk of damage to adjacent tissues. However, its use may be limited in cases where patient anatomy does not allow stable placement of the patient tool (PT) or in instances where the target lesion is not well visualized on CT imaging. Contraindications may include severe patient movement disorders, metallic implants causing imaging artifacts, or insufficient space for the PT placement near the target area [3,4].

The findings of this study provide compelling evidence for the efficacy of the SIRIO navigation system in interventional procedures for vertebral RFA with neuroprotection, particularly in the treatment of primary and metastatic bone tumors. Our comprehensive

analysis demonstrated significant improvements in procedural efficacy and patient outcomes when the SIRIO system was utilized. SIRIO-assisted neuroprotection is associated with a significant reduction in both dose-length product (DLP) and procedural time. The average DLP was significantly lower in the SIRIO group (307.42 mGycm) compared to the non-SIRIO group (460.31 mGycm). Reducing radiation exposure to patients is clinically relevant, as it minimizes potential risks associated with radiation, in line with existing literature highlighting the importance of reducing radiation doses in interventional radiology [6]. According to many studies, the majority of the radiation dose is contributed by the preliminary planning CT [8–10], and it is precisely at this stage that the navigation system facilitates the procedure. This data correlates with the procedural time, which was significantly shorter in the SIRIO-assisted group, with an average of 13.48 min compared to 32.26 min in the unassisted group. The navigation system helped to identify the most direct and shortest path of the needle to reach the epidural space (Figure 7), also making the procedure quicker. This evidence is significant, as shorter procedural times can reduce the risk of complications and improve overall patient performance in clinical settings [11]. The efficacy improvements observed with the use of the SIRIO navigation system highlight its potential to improve workflow and optimize resource utilization in interventional oncology.

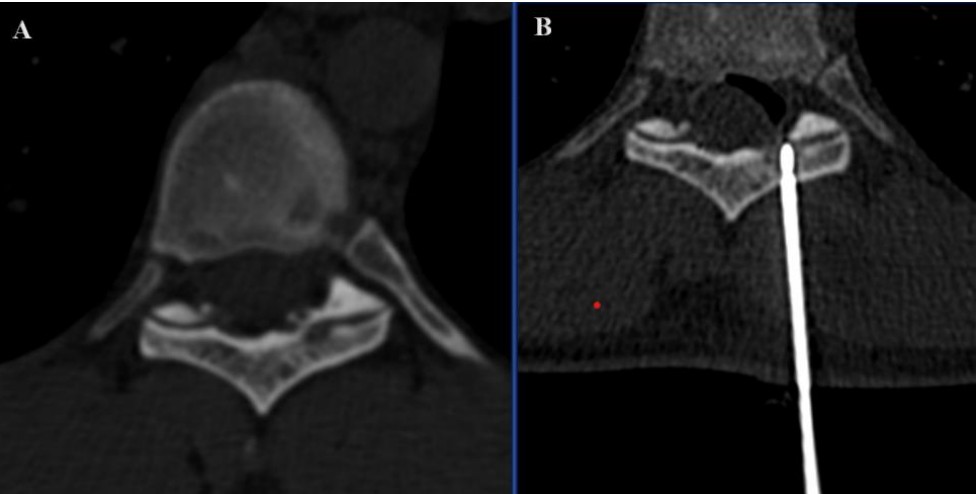

**Figure 7.** Axial (**A**) image demonstrating the presence of an osteoid osteoma located in adjacency to the inferior articular facet of D9. Axial (**B**) image demonstrating correct needle positioning and $CO_2$ epidural dissection.

The integration of neuroprotection techniques with the SIRIO navigation system has further improved patient outcomes. The precise targeting facilitated by SIRIO ensured effective neuroprotection, which is critical for minimizing nerve damage and reducing post-procedural pain. This combination of advanced navigation and neuroprotection techniques resulted in substantial pain reduction, as measured by the Visual Analog Scale (VAS) at procedural time (VAS Time 0) and three months post-procedure (VAS Time 1). This also resulted in a dosage reduction in hypnotics and opioids administered for sedation during LTA. This finding aligns with previous studies that have demonstrated the benefits of advanced navigation systems in enhancing the accuracy and efficacy of interventional procedures [12–16]. Moreover, both SIRIO- and non-SIRIO-achieved neuroprotection showed excellent technical success and the absence of complications, underscoring the reliability of these methods in clinical practice. Our subgroup analysis, which examined differences in VAS Time 1 across different age groups, indicated significant variations in pain outcomes, particularly in the non-SIRIO-assisted procedures. Younger patients (<50 years) experienced less pain at three months post-procedure compared to older patients, highlighting the need for tailored pain management strategies in different age cohorts.

The Spearman correlation analysis between tumor type and VAS Time 1 showed a very weak positive correlation, indicating that tumor type does not significantly influence pain outcomes. This suggests that the benefits of SIRIO-assisted neuroprotection in reducing procedural pain are consistent across various tumor types, reinforcing its versatility and effectiveness in diverse clinical scenarios.

Despite the promising findings, this study has several limitations. The sample size was relatively small, particularly when subdivided into SIRIO-assisted and non-assisted groups, which may limit the generalizability of the results. Additionally, the retrospective nature of the study may introduce selection bias, and the lack of randomization could affect the validity of the comparisons. Lastly, the study did not include a long-term follow-up to assess the durability of the procedural outcomes and the potential for late complications.

As treatment targets become more complex and patients' fragility increases, neuroprotection is becoming increasingly relevant in the interventional oncology field [2,17]. This study demonstrated that the dissection of the epidural space for the treatment of primary benign and metastatic lesions of the vertebral bones is technically feasible and facilitated by the use of a CT-guided infrared augmented reality navigation system.

## 6. Conclusions

In conclusion, this study provides compelling evidence that the SIRIO navigation system significantly reduces both the radiation dose (DLP) and procedural time in neuroprotection procedures. The benefits of the SIRIO system are consistent across different patient demographics and tumor characteristics, as demonstrated by our results. These findings suggest that integrating advanced navigation systems like SIRIO into interventional oncology practice can lead to substantial improvements in patient care and procedural efficacy. Future studies with larger, randomized cohorts are warranted to further validate these results and explore the long-term benefits of the SIRIO system in various clinical settings.

**Author Contributions:** Formal analysis, M.P.; Writing—review and editing, R.C.; Visualization, G.P., C.A., E.V., A.B., B.B.Z. and R.F.G.; Supervision, E.F. All authors have read and agreed to the published version of the manuscript.

**Funding:** This research received no external funding.

**Institutional Review Board Statement:** The study was conducted in accordance with the Declaration of Helsinki and approved by the Institutional Review Board of Campus Bio-Medico University Hospital (Prot. No. 01/19 OSS Com Et CBM).

**Informed Consent Statement:** Informed consent was obtained from all subjects involved in the study.

**Data Availability Statement:** The data presented in this study are available on request from the corresponding author.

**Conflicts of Interest:** The authors declare no conflicts of interest.

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
