# Peer review of "Augmented Reality Navigation System (SIRIO) for Neuroprotection in Vertebral Tumoral Ablation"

_curroncol, doi:10.3390/curroncol31090376_

Round 1
Reviewer 1 Report
Comments and Suggestions for Authors
This article is a study on the application of augmented reality navigation system (SIRIO) in spinal lesion ablation surgery. The primary findings indicate that surgeries augmented with the SIRIO system demonstrated significant benefits, particularly in reducing radiation exposure, measured by the dose-length product (DLP), and expediting surgical time. Moreover, both the SIRIO-assisted and non-assisted groups exhibited commendable clinical outcomes, as evidenced by high rates of technical success, low complication rates, and satisfactory pain scores assessed via the Visual Analog Scale (VAS). The conclusions drawn from this study underscore the promising capacity of the SIRIO system to bolster the safety and precision of surgical interventions.
The SIRIO system's precise navigation capabilities are laudable as they mitigate potential harm to adjacent nerve structures, thereby enhancing surgical safety and curtailing both radiation exposure and operative duration. Nonetheless, the article does raise several points for further consideration:
1. The methodology section does not clarify whether the study groups were assigned through a randomized process, a detail that could significantly influence the reliability of the comparative analyses presented.
2. The article's omission of the SIRIO system's operational details is notable. It would be beneficial to include a concise overview of the system, a thorough description of its operational protocols, and illustrative figures to elucidate the procedural steps.
3. While the SIRIO system's benefits are evident, every technology has its specific applications and limitations. An expanded discussion on the system's indications and contraindications would provide a more comprehensive understanding of its clinical utility.
I extend my appreciation for the dedication to this research endeavor and for the valuable contributions to the domain of spinal lesion ablation surgery.
Author Response
Comment 1: The methodology section does not clarify whether the study groups were assigned through a randomized process, a detail that could significantly influence the reliability of the comparative analyses presented.
Response 1: Thank you for your insightful feedback. We have revised the methodology section to explicitly state that this is a non-randomized, retrospective study. This clarification highlights the study design and addresses the potential limitation you identified regarding group assignment.
Comment 2: The article's omission of the SIRIO system's operational details is notable. It would be beneficial to include a concise overview of the system, a thorough description of its operational protocols, and illustrative figures to elucidate the procedural steps.
Response 2: Thank you for your suggestion to include a detailed explanation of the SIRIO system's operation. We would like to point out that the introduction already contains a paragraph that succinctly explains the functioning of the SIRIO system. This paragraph describes the key components and the operational process, including the reconstruction of a 3D model from CT images, the role of the patient and needle tools, and the real-time guidance provided by the system. Given that this explanation is already present and effectively communicates the system's operation, we have chosen to leave this section unchanged.
Comment 3: While the SIRIO system's benefits are evident, every technology has its specific applications and limitations. An expanded discussion on the system's indications and contraindications would provide a more comprehensive understanding of its clinical utility.
Response 3: Thank you for your thoughtful suggestion. We agree that discussing the specific applications and limitations of the SIRIO system would enhance the manuscript. We expanded the discussion section to include a more detailed analysis of the system's indications and contraindications, providing a broader perspective on its clinical utility.
The revisions clarify the study design, provide detailed information on the SIRIO system, and expand on its clinical applications and limitations. I believe these improvements enhance the manuscript's clarity and rigor. I remain available for any further modifications if needed.
Reviewer 2 Report
Comments and Suggestions for Authors
The study is so interesting, however, I have some concerns to discuss.
- Methodology and Sample Size:
- How was the sample size of 28 vertebral RTA procedures determined, and what statistical power does this sample size provide for detecting differences between SIRIO-assisted and Non-SIRIO-assisted procedures?
- Outcome Measures:
- The study uses dose-length product (DLP) and procedural time as primary outcomes. Are these the most appropriate primary outcomes for assessing the efficacy and safety of the SIRIO system? Should other metrics, such as long-term clinical outcomes or patient-reported outcomes, be considered as primary endpoints?
- Statistical Analysis:
- The study employs multiple regression analysis to assess the impact of various predictors on DLP and procedural time. Were all relevant confounding variables included in the regression models, and how was multicollinearity addressed?
- Comparative Efficacy:
- The results show significant reductions in DLP and procedural time with SIRIO assistance. However, there was no significant difference in pain scores (VAS Time 1) between the groups. How do the authors interpret the clinical significance of these findings, and what implications do they have for the adoption of the SIRIO system in clinical practice?
- Future Research Directions:
- The authors conclude that further studies with larger, randomized cohorts are needed. What specific aspects of the SIRIO system's efficacy and safety should future research focus on, and what study designs would be most appropriate to address these aspects?
- Can this method be used in the presence of a tumor? Please refer to the following literature for discussion. Comprehensive treatment outcomes of giant cell tumor of the spine: A retrospective study. Medicine (Baltimore). 2022;101(32):e29963. doi:10.1097/MD.0000000000029963
Author Response
Comment 1: How was the sample size of 28 vertebral RTA procedures determined, and what statistical power does this sample size provide for detecting differences between SIRIO-assisted and Non-SIRIO-assisted procedures?
Response 1: The sample size of 28 vertebral RTA procedures was determined based on the availability of eligible cases during the study period, rather than through a formal power calculation. As this was a retrospective analysis, we included all consecutive procedures that met the inclusion criteria. We acknowledge that the sample size may limit the statistical power to detect differences between the SIRIO-assisted and Non-SIRIO-assisted groups. However, despite this limitation, the study did reveal statistically significant differences in key outcomes, suggesting a meaningful impact of the SIRIO system. Future studies with larger sample sizes and prospective designs are needed to confirm these findings and provide more robust statistical power.
Comment 2: The study uses dose-length product (DLP) and procedural time as primary outcomes. Are these the most appropriate primary outcomes for assessing the efficacy and safety of the SIRIO system? Should other metrics, such as long-term clinical outcomes or patient-reported outcomes, be considered as primary endpoints?
Response 2: We agree that long-term clinical outcomes and patient-reported outcomes are important metrics to consider. However, our primary goal in this study was to assess the immediate procedural efficacy and safety of the SIRIO system. We chose dose-length product (DLP) and procedural time as primary outcomes because they directly reflect the system's ability to reduce radiation exposure and enhance procedural efficiency—key factors in ensuring patient safety. Our data suggest that the SIRIO system not only maintains clinical safety but also provides additional benefits by making the procedure quicker and reducing radiation exposure, which are crucial for patient well-being. We acknowledge that further studies should incorporate long-term and patient-reported outcomes to provide a more comprehensive evaluation of the system's overall impact.
Comment 3: The study employs multiple regression analysis to assess the impact of various predictors on DLP and procedural time. Were all relevant confounding variables included in the regression models, and how was multicollinearity addressed?
Response 3: Thank you for your detailed question regarding the statistical analysis. In our multiple regression analysis, we included key variables such as age, tumor type, and SIRIO assistance to assess their impact on DLP and procedural time. We carefully selected these predictors based on their clinical relevance and potential influence on the outcomes. To address multicollinearity, we assessed the Variance Inflation Factor (VIF) for each predictor in the regression models. All VIF values were within acceptable limits, indicating that multicollinearity was not a significant issue in our analysis. While we believe we included the most relevant confounding variables, we acknowledge that other factors might also contribute to the outcomes, and future studies could explore additional variables.
Comment 4: The results show significant reductions in DLP and procedural time with SIRIO assistance. However, there was no significant difference in pain scores (VAS Time 1) between the groups. How do the authors interpret the clinical significance of these findings, and what implications do they have for the adoption of the SIRIO system in clinical practice?
Response 4: The significant reductions in DLP and procedural time with SIRIO assistance indicate that the system enhances procedural efficiency and safety by reducing radiation exposure and shortening the duration of the procedure. These benefits are particularly important in minimizing patient risk and improving the overall workflow in clinical settings. Regarding the lack of significant difference in pain scores (VAS Time 1) between the groups, we interpret this as an indication that the SIRIO system maintains the same level of clinical efficacy in terms of pain management as the standard approach, while offering additional advantages in terms of efficiency and safety. This suggests that the adoption of the SIRIO system could provide meaningful improvements in clinical practice, particularly in environments where reducing procedural time and radiation exposure is a priority.
Comment 5: The authors conclude that further studies with larger, randomized cohorts are needed. What specific aspects of the SIRIO system's efficacy and safety should future research focus on, and what study designs would be most appropriate to address these aspects? Can this method be used in the presence of a tumor? Please refer to the following literature for discussion. Comprehensive treatment outcomes of giant cell tumor of the spine: A retrospective study. Medicine (Baltimore). 2022;101(32):e29963. doi:10.1097/MD.0000000000029963
Response 5: We agree that further studies are essential to validate and expand on our findings. Future research should focus on the long-term outcomes of using the SIRIO system, including its impact on tumor control, recurrence rates, and patient-reported quality of life. Additionally, investigating its efficacy and safety in more complex cases, such as larger or more challenging tumors, would be valuable.Randomized controlled trials with larger cohorts would be the most appropriate study design to address these aspects, as they would help eliminate potential biases and provide robust evidence for the SIRIO system's benefits. Including comparisons with other advanced navigation systems would also be beneficial to establish its relative effectiveness. Regarding the application of this method in the presence of other complex tumor lesions, the SIRIO system is particularly suited for precise needle placement, which is crucial when treating tumors near critical structures. Given that the article already demonstrates the SIRIO system's effectiveness in precise probe placement for complex spine tumor cases with highly promising results, we do not find it appropriate to introduce a comparison with a lesion type not included in our study's database. However, we remain available to include this information in the discussion if you deem it necessary.
The revisions clarify the study design, provide detailed information on the SIRIO system, and expand on its clinical applications and limitations. We believe these improvements enhance the manuscript's clarity and rigor, and remain available for any further modifications if needed.
Round 2
Reviewer 2 Report
Comments and Suggestions for Authors
The authors replied well, so the manuscript suitable for publication.